# The Three-Dimensional Reference Interaction Site Model Approach as a Promising Tool for Studying Hydrated Viruses and Their Complexes with Ligands

**DOI:** 10.3390/ijms25073697

**Published:** 2024-03-26

**Authors:** Marina V. Fedotova, Gennady N. Chuev

**Affiliations:** 1G.A. Krestov Institute of Solution Chemistry, The Russian Academy of Sciences, Akademicheskaya St., 1, 153045 Ivanovo, Russia; 2Institute of Theoretical and Experimental Biophysics, The Russian Academy of Sciences, Institutskaya St., 142290 Pushchino, Russia

**Keywords:** 3D-RISM theory, viruses, solvation effects, virus–host cell interactions, protein–ligand binding

## Abstract

Viruses are the most numerous biological form living in any ecosystem. Viral diseases affect not only people but also representatives of fauna and flora. The latest pandemic has shown how important it is for the scientific community to respond quickly to the challenge, including critically assessing the viral threat and developing appropriate measures to counter this threat. Scientists around the world are making enormous efforts to solve these problems. In silico methods, which allow quite rapid obtention of, in many cases, accurate information in this field, are effective tools for the description of various aspects of virus activity, including virus–host cell interactions, and, thus, can provide a molecular insight into the mechanism of virus functioning. The three-dimensional reference interaction site model (3D-RISM) seems to be one of the most effective and inexpensive methods to compute hydrated viruses, since the method allows us to provide efficient calculations of hydrated viruses, remaining all molecular details of the liquid environment and virus structure. The pandemic challenge has resulted in a fast increase in the number of 3D-RISM calculations devoted to hydrated viruses. To provide readers with a summary of this literature, we present a systematic overview of the 3D-RISM calculations, covering the period since 2010. We discuss various biophysical aspects of the 3D-RISM results and demonstrate capabilities, limitations, achievements, and prospects of the method using examples of viruses such as influenza, hepatitis, and SARS-CoV-2 viruses.

## 1. Introduction

Viruses are unique small infectious agents that require host cells to replicate their genetic material; therefore, they cannot reproduce and carry out metabolic processes without a host cell. As obligate intracellular parasites, they depend on the host cell for almost all of their life-sustaining functions. After gaining access to the specific host tissue, target-cell infection is achieved at the initial stage of virus replication when the virus recognizes and binds to a receptor on the target-cell surface [1]. To date, much is known about the functioning of viruses, but the molecular mechanisms of virus infection of hosts are not fully understood, although the latest pandemic had a huge impact on scientific research, especially in virology. Some of the experimental tools used to study the structure and properties of viruses are centrifugation, serological studies, tissue culture techniques, electron microscopy, atomic force microscopy, differential scanning fluorimetry, X-ray diffraction, etc. However, as rightly noted in [2], “due to the liquid, lipid environment and the dynamic nature of the virus–host cell interactions, detailed structures of the binding partners and the progression of molecular events leading to entry are difficult to obtain experimentally”. The unique nature of these agents, as well as the fast evolution of new viral strains, requires new experimental and theoretical techniques. The results of the last coronavirus pandemic and the post-pandemic period, during which the number of in silico studies of viruses has increased significantly (manifold), show that non-empirical methods are effective tools for this field due to their low cost, as well as their freedom from safety and ethical restrictions [3]. Like any other study, computational studies are aimed at understanding the mechanism of viral infection as it relates to the development of new small-molecule drugs that inhibit virus–host cell binding. A wide range of approaches are actively involved in non-empirical studies of viruses—from a bioinformatics analysis (see, for instance, [4,5]) to coarse-grained computational modeling [6] and long-timescale all-atom MD simulations (see, for instance, [7,8,9,10]) as well as a fashionable machine-learning-based clustering technique [11] including deep learning [12,13]. A detailed review of various computational methods and their combinations can be found in [3], where such aspects as molecular biophysics, bioinformatics, cheminformatics, machine learning, and mathematics are discussed. However, the authors of this paper missed one non-empirical method for studying viruses: the 3D-RISM (three-dimensional reference interaction site model) approach [14,15,16] of the statistical theory of liquids, which can be considered a simplified version of the classical site density functional theory (SDFT) [17,18,19,20]. In the present review, we fill this gap, including consideration of our recent contribution [5] to this topic.

## 2. The 3D-RISM Approach as a Promising Tool for Studying Hydrated Biomolecules

In recent years, the 3D-RISM method, also known as the molecular theory of solvation, including its combination with various predictive models [21,22,23], has become increasingly popular in studies of both small [24,25,26,27,28,29,30,31,32] and large [33,34,35,36,37,38,39,40] biomolecules, including viruses [5,41,42,43,44,45]. This modern approach has achieved great success in exploring the diversity of processes occurring in biosystems, such as solvation, molecular recognition, including binding with ligands, and protein–protein interactions (see, for instance, [33,34,35,36,37]). The 3D-RISM method, first of all, is capable of providing information on the equilibrium solvation structure, thermodynamics, and energetics as well as on features of complex formation for biocompounds in solution, although its potential is much wider and can be used to study the physicochemical behavior of solutes of various chemical natures such as small organic molecules [46,47,48,49,50,51], ionic liquids [52,53,54], polyelectrolyte gels [55], nanotubes [56,57,58], etc. (see also the references in reviews [59,60]). The discussed theory properly takes into account the chemical nature of compounds and all the basic interactions between their various moieties, allowing the reproduction of key solvation features such as H-bonding, solvent structural state, hydrophobic interactions, etc. [59,61]. All of the above allow us to talk about this theory as a powerful tool for a description of the solvation effects. From a practical perspective, this approach may be useful for potential drug screening and virulence assays [44]. In contrast to MD simulations with explicit solvent models that are time-consuming when studying large hydrated biomolecules, the 3D-RISM approach allows direct, rapid, and fairly accurate prediction of the spatial solvent distribution around a solute with relatively low computational costs. Moreover, although MD simulations can study the behavior of biomolecules in full atomic detail and at very fine temporal resolution, in the case of viruses, they cannot reproduce all the details of the binding of the virus with the host cell due to high computational costs and the need for a large conformational space [62]. To date, the 3D-RISM method has been implemented in a number of software packages in the field of computational chemistry and materials, such as ADF [63], AMBER [64], AutoDock [65], EPISOL [66], MOE [67], NWChem [68,69], Quantum Espresso [70], etc., and is thus available to researchers.

The 3D-RISM theory is based on the integral equation formalism using the Ornstein–Zernike-type integral equations for molecular liquids (3D-RISM integral equations) [16,71,72,73,74]. It yields a detailed molecular picture of the solvation of a molecule with an arbitrary shape and size on the molecular–atom level, based on the solute–solvent interactions by the spatial distribution functions (SDFs). These functions are the result of the numerical solution of the 3D-RISM Ornstein–Zernike (OZ) equation with the appropriate closure relation. As a rule, the 3D Kovalenko–Hirata (KH) closure [75,76,77,78] is used in studies of biomolecular solvation. Using the 3D-RISM method, the solvation structure and solvation thermodynamics can be described. For example, one can obtain a visualization of the hydration layer around a solute molecule, the total hydration number of the solute molecule, the solvation free energy of a solute in a multicomponent solvent, the potential of mean force that, in particular, estimates the site binding affinity of solvent species to the solute molecule [36,59,79], the partial molar volume of a solute molecule [80,81], and bulk solvent compressibility [82] (for more details, see Section 4 with methodology and corresponding Equations (4)–(8)).

At the same time, it is known (see, for instance, the discussion in Refs. [39,83]) that the use of 3D-RISM theory can give an incorrect estimate of thermodynamic quantities, including the solvation free energy compared to the experimental one. Therefore, to increase the accuracy of the 3D-RISM solvation energies, a number of corrections, such as the Gaussian fluctuation (GF) approximation [84], advanced pressure correction (PC+) [85], or universal correction (UC) [86], using the GF approximation corrected with partial molar volume, have been proposed.

It should be noted that the 3D-RISM method is used as a standalone tool for studying the structure and thermodynamics of biosystems, as well as in combination with other computational chemistry approaches such as DFT, MD, QM, MM, docking, and structural bioinformatics to ensure additional information about the properties of a system and gain a comprehensive understanding of its behavior. In particular, as the binding free energy, Δ*G*_bind_, of a protein–ligand complex in the solvent is defined, first of all, by the solvation free energy of participants of the binding process, the 3D-RISM calculations for Δ*G*^solv^ are coupled with most of the above techniques for an estimation of the binding affinity of a ligand (a pharmaceutical drug) to a target protein.

## 3. Applications of 3D-RISM Theory for Studying Viruses

Although 3D-RISM is an attractive tool for studying macromolecules, to date, there are only a few studies focusing on viruses.

### 3.1. Influenza Virus

One of the first papers on this topic was a paper on proton transport through the influenza A M2 channel [41]. It should be noted that the M2 protein channel located in the lipid envelope of the virus plays an important role in proton transport and replication of the H1N1 influenza A virus [87,88] and is, therefore, considered as a target for the development of antiviral drugs (for example, amantadines) [89]. The M2 channel is highly selective for protons, and its gating is controlled by pH. A key suggestion in its gating mechanism is that its amino acid residue His37 may act as a pH sensor switch, turning the gate on/off, and as a selective filter, allowing only protons to pass through, but not other cations [41,90]. Numerous efforts were aimed at understanding the mechanism of proton transport across the virus membrane as one of the central points in drug design. However, this problem is still not fully understood due to the annual emergence of new strains of the virus. To clarify the molecular mechanism of gating and proton conduction in the channel, the authors [41] have performed the 3D-RISM-KH calculations and analyzed the 3D distribution and energetic barriers for water molecules as well as hydronium ions, a model of protons, inside the M2 channel (PDB id: 1NYJ) with five different protonated states: PS (non-PS (0H), single PS (1H), double PS at diagonal position (2H), triple PS (3H), and quadruple PS (4H)). In other words, these protonated states are distinguished by the number of protonated histidines (His37) in the gating region, from no histidine protonated (0H) to four histidines protonated (4H) [41].

The obtained SDFs (Figure 1) clearly demonstrate the increasing accessibility of water to the channel pore with a change in the protonated state of the channel from 0H to 4H in the set of 0H < 1H < 2H < 3H < 4H. The authors explained this result by an increase in the pore diameter due to electrostatic repulsion among the protonated His37 [41]. The analysis of these functions allowed the authors to identify two conformations of the channel. 

The first is the “closed” form of the channel in the 0H, 1H, and 2H states, since water distributions are not detected in the regions of His37 and the region with the Trp41 residues. In this case, the PMFs calculated by Equation (5) in Section 4 have high positive energetic barriers that water cannot overcome due to steric hindrance caused by both pores and solvent molecules. The situation is the same for hydronium ions, but with a lower distribution and higher barrier in PMFs compared to those of water, which indicates the impossibility of distributing hydronium ions (or protons) in the channel.

The second conformation is the “open” form of the channel in the 3H and 4H states with a continuous distribution of solvent along the pore. In this case, the PMFs are negative along the whole channel pore, demonstrating the ability of water molecules to permeate through the channel. As to H_3_O^+^ ions with relation to the 3H and 4H forms of the channel, the behavior of their PMFs is not negative; however, the barrier heights in PMFs are only 2–3 and 5–7 kJ/mol, which is comparable to the thermal energy. Thus, protons can overcome thermal fluctuations in protein conformation. It is interesting that the barrier for the 3H form is lower than for the 4H form. Such a behavior was explained by the authors as a result of the competition between two factors when the protonation level of the channel increases: “enhanced pore diameter due to the increased coulomb repulsions among the protonated histidines, and enhanced coulomb repulsion between the protonated histidines and hydronium ions” [41]. The balance between these factors is observed in the 3H form of the M2 channel.

Based on the data, the authors have proposed [41] a novel model for a proton-transfer mechanism in the influenza A M2 channel, which is consistent with the conclusions drawn from experimental data [90]. For the mechanism to work, two histidines are required—in protonated and non-protonated states. According to the model, the process of proton transfer from a water molecule to another water molecule is as follows: “…a hydronium ion hands a proton to a non-protonated His through a hydrogen bond between them, and then a protonated His releases a proton to a water molecule via a hydrogen bond.” [41].

Another study of the influenza virus [43] was carried out by almost the same team of authors. The authors investigated the binding affinity of the drug oseltamivir to the wild-type of viral influenza B neuraminidase and to its resistance-associated mutant forms. Neuraminidase (NA) is one of two glycoproteins located on the surface membrane of the influenza B virus. It is involved in virus replication by splitting the sialic acid residues (Sias) and is responsible for releasing new virions from the infected host cell to infect new cells. Oseltamivir is one of the FDA (U.S. Food and Drug Administration)-approved and the WHO (World Health Organization)-recommended NA inhibitors that is widely used to prevent and treat both influenza A and B viruses (see, for instance, [91,92]). Because NA is used by influenza virions to exit the cell, oseltamivir, like other NA inhibitors (such as zanamivir, peramivir, etc.), should prevent the release of new viral particles in a living body [93]. However, oseltamivir’s risk–benefit ratio is controversial (see, for instance, [94]), since among circulating influenza viruses, strains resistant to oseltamivir have been identified [95], and the clinical efficiency of this drug still remains unproven [96]. Nevertheless, research into its efficiency in treating influenza has been ongoing since 1999, when this drug was approved by the FDA.

The main goal of the paper [43] was to understand at the molecular level the oseltamivir efficiency toward wild-type neuraminidase as well as its variants with three single substitutions E119G, R152K, and D198N in terms of molecular recognition. To clarify the mechanism of receptor inhibition by a drug, it is important to know the binding affinity of the ligand to the receptor. To estimate the binding affinity, the authors have used the 3D-RISM-KH approach in combination with molecular mechanics (MM)—the so-called MM/3D-RISM calculation [61].

Upon the protein–ligand binding occurring in aqueous media, the hydration/dehydration of the compounds involved in complex formation plays one of the key roles under this process. Therefore, as the first step of the study, the authors [43] have analyzed the changes in water distribution upon the binding of oseltamivir (PDB id: 2HU4) with wild-type NA (PDB id: 1NSC). To this end, they used the 3D-RISM-KH SDFs of water oxygens and hydrogens around the ligand (Figure 2) and residues in the active site of the receptor (Figure 3) before and after the complex formation, as well as the corresponding radial distribution functions. The structures of free oseltamivir and NA were found to be stabilized by the hydrating water, confirming the known hydration effect on protein stability. Upon complex formation, both compounds dehydrate, as evidenced by a decrease in the distribution of water around them. Moreover, after binding, some of the water molecules in the NA binding pocket were replaced by a ligand. At the same time, some portion of the solvent remains in the pocket as the bridging water molecules between the residue and the ligand [43].

In the next step, the authors [43] estimated the binding affinity of oseltamivir to three NA mutants, i.e., the protein–ligand binding free energy (Δ*G*_bind_). In the MM/3D-RISM approach, Δ*G*_bind_ is calculated according to the standard thermodynamic cycle and is the sum of the contribution from solute molecules (∆*G*_bind_^MM^) and solvation free energy (∆*G*^solv^), where ∆*G*_bind_^MM^ bind is a combination of molecular mechanics energy (∆*E*^MM^) and structural entropy (*T*∆*S*^MM^) of protein, i.e.,
Δ*G*_bind_ = ∆*E*^MM^ + ∆*G*^solv^ − *T*∆*S*^MM^
where ∆*G*^solv^ consists of polar and nonpolar parts, ∆*G*^solv^ = ∆*G*^pol^ + ∆*G*^nonpol^. The solvation free energy is traditionally estimated by the Poisson–Boltzmann [21] and Generalized Born solvent accessible area [22] (PBSA and GBSA, respectively) continuum solvation models. The MM/3D-RISM-KH results are obtained by replacing the PB or GB polar and SASA nonpolar solvation terms in the above equation by the solvation free energy from Equation (6) (see Section 4). The latter is obtained by solving the 3D-RISM-KH integral Equations (1) and (3) (see Section 4) at the selected arrangements of the biomolecule for snapshots taken along the MD trajectory, with water molecules stripped off [61].

In the framework of the MM/3D-RISM-KH methodology, the energetics related to the binding affinity of oseltamivir to strains was compared with the Δ*G*_bind_ and its components for the corresponding complex with wild-type NA as well as with experimental data for mutated systems. These results indicated that the ligand–receptor binding free energy is determined “by a subtle balance of two major contributions that largely cancel out each other: the ligand–receptor interactions and the dehydration free energy” [43]. In addition, the dehydration free energy is governed by the change in the water distribution at the active-site pocket before and after the inhibitor binding. As was shown in [43], the data for the relative Δ*G*_bind_ reproduce the experimental tendency in the resistivity, namely, the different levels of resistance of the mutants to oseltamivir. The authors have determined the high-level resistance of the E119G and R152K strains as a result of a reduction in the direct drug–target interaction as well as the low-level resistance of the D198N strain when this reduction is absent.

**Figure 3 ijms-25-03697-f003:**
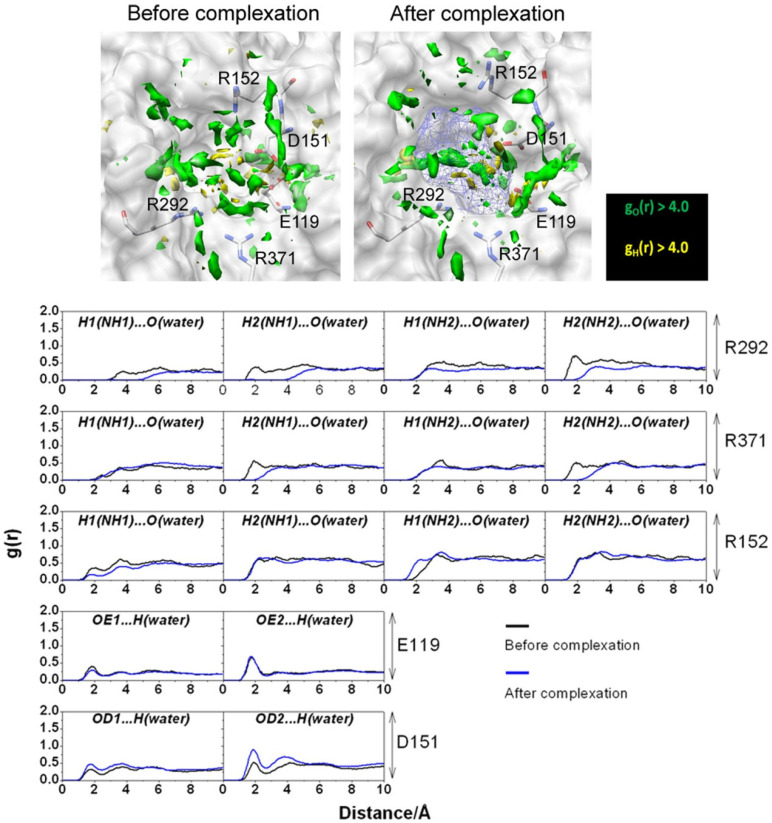
(**Upper**) SDFs of water O (green) and H atoms (yellow) with g(**r**) > 4.0 within 7.0 Å of oseltamivir (depicted as blue mesh sphere). (**Lower**) The 3D-RISM RDF of hydrogen-bonded pairs between water molecules and five amino acid residues before and after complexation. Reprinted from Phanich et al. [43], Copyright (2016), with permission from John Wiley & Sons, Inc.

### 3.2. Hepatitis Virus

The paper [42] is devoted to the assessment of metal-assisted nucleophile activation in the hepatitis delta virus (HDV) ribozyme (PDB id: 3NKB). The calculations have been performed using molecular modeling (QM/MM (quantum mechanical/molecular mechanical) MD simulations) and the combined MM/3D-RISM method. The HDV ribozyme (HDVr) is a non-coding RNA found in the HDV that is required for viral replication. It is the sole human virus that uses ribozyme activity to infect its host cells [97]. This process is facilitated by HDVr catalysis; therefore, the catalytic activity of HDVr is actively discussed in the literature. The HDVr catalyzes RNA backbone cleavage, and the 2′-hydroxyl group adjacent to the cleavage site phosphate group is the nucleophile in a backbone cleavage reaction [98]. The HDVr is able to utilize various catalytic strategies, such as site-specific shifts of nucleobase p*K*_a_’s and recruitment of metal ions. Regarding HDVr catalysis, the latter is under close scrutiny. In particular, molar concentrations of monovalent ions [98] or millimolar concentrations of divalent ions [98,99] can support catalysis to achieve optimal reaction rates under near-physiological conditions. According to crystallographic data [100], there is a metal-binding pocket in the HDVr active site, and thus, the metal ion can interact directly with the scissile phosphate and the nucleophile. As stated in [100], an ion, such as Mg^2+^, “can serve as a Lewis acid, facilitating deprotonation of the nucleophile and stabilizing the conformation of the cleavage site for in-line attack of the nucleophile at the scissile phosphate”. Although it is already known that the divalent metal ions are more effective in supporting catalysis and play a specific role played in the catalytic mechanism, our understanding of ribozyme self-cleavage remains incomplete.

The paper [42] provides a detailed computational perspective on active-site Mg^2+^ association and its contribution to catalysis by studying divalent metal ion binding and nucleophile activation. The QM/MM method was used to simulate two reaction channels in the HDVr involving a site-bound Mg^2+^ or Na^+^ and to map the free-energy surfaces with different active-site ion occupancies. These results have shown that a possible pathway with metal-assisted nucleophile activation is based on the rate-controlling transition state barrier departing from the supposed metal-bound active state [42]. The 3D-RISM and MM/3D-RISM techniques were applied to assess energetics and solvation effects in the system. In particular, the 3D-RISM distribution of Na^+^ ions around HDVr is consistent with the corresponding distribution obtained from MD simulations when the buried Na^+^ is located in the vicinity of a nucleophile (Figure 4).

Then, using a thermodynamic cycle describing the protonation and binding of Mg^2+^ ions and taking into account changes in the background Na^+^ concentration, the authors [42] obtained the relative free energies. Let us remind ourselves once again that under the MM/3D-RISM method, the PBSA solvation model in the MM/PBSA analysis of the thermodynamics of the MD trajectories is replaced by the solvation free energies calculated in the framework of the 3D-RISM-KH theory [61]. As p*K*_a_ shifts for deprotonation of the nucleophile, it is possible to determine p*K*_a_ in proportion to the difference in the deprotonation free energies or the binding free energies, as well as in relation to the change in ionic strength. Following this path, the authors [42] have obtained the MM/3D-RISM results for p*K*_a_ shifts and relative deprotonation free energies (Δ*G*^Mg2+^ − Δ*G*^Na+^) under varying background salt species and concentrations. The calculations give p*K*_a_ of ∼4 units in the presence of Mg^2+^ ions (background salt concentration is 20 mM) as well as ∼2.5 units in the presence of Na^+^ ions (background salt concentration is 1 M). The method also provides a free-energy shift of ∼2 kcal/mol toward the Mg^2+^ bound state, i.e., in favor of the Mg^2+^ bound mechanism. However, as the authors note, the calculated value is quite far from the experimental value, equal to ∼16 kcal/mol [101]. Nevertheless, the results presented in [42] confirm the fact that the Mg^2+^ ion participates in the cleavage reaction of the HDV ribozyme under biologically relevant buffer conditions.

### 3.3. SARS-CoV-2 Virus

As already mentioned, the COVID-19 pandemic has sparked a new wave of interest in the study of viruses. Starting in 2021, a number of 3D-RISM articles have been devoted to the study of the SARS-CoV-2 virus.

According to phylogenetic analysis, the coronaviruses, including human ones such as the previous severe acute respiratory syndrome coronavirus (SARS-CoV) and novel severe acute respiratory syndrome coronavirus-2 (SARS-CoV-2) or Middle East respiratory syndrome coronavirus (MERS-CoV), reveal high similarity [102]. They belong to the Betacoronaviruses genus and are enveloped, positive-strand RNA viruses [103]. The life cycle of any virus, including that of coronaviruses, consists of three stages: viral entry, replication, and release (shedding). In the case of SARS-CoV-2, a causative agent of COVID-19, its entry into human host cells is mediated by the SARS-CoV-2 spike (S) glycoprotein. With the use of S protein, the virus interacts with the human receptor membrane protein angiotensin-converting enzyme 2 (hACE2). Note that hACE2 is highly expressed in various organs, such as the lungs, heart, kidneys, and intestine [104]. S protein, protruded from the virion surface, is made of two subunits, S1 and S2, responsible for receptor recognition and membrane fusion, which are essential steps for viral entry into the host cell [105]. The S1 subunit comprises the receptor-binding domain (RBD), which binds with the peptidase domain (PD) of the hACE2 receptor on the host cell (Figure 5), while the S2 subunit provides membrane fusion [106]. The RBD selectively recognizes the hACE2 and contributes to the stabilization of the prefusion conformation state. In turn, the binding of hACE2 to the RBD triggers various processes of invasion and proliferation. Thus, the binding of S proteins to the hACE2 receptor is an essential initial process of SARS-CoV-2 infection. Moreover, the solvent effects play one of the key roles in this protein–protein binding. Therefore, it is not surprising that these aspects have been the subject of 3D-RISM research in several works [5,44,45].

Kobryn et al. [44] investigated the role of solvation in RBD–hACE2 binding. To this end, they considered a system where the solute is the SARS-CoV-2–hACE2 complex, while the solvent contains water molecules, ions (Na^+^ and Cl^−^), and dissolved drug-like molecules (small organic molecules of the types of N-acetyl-*D*-glucosamine, NAG). The RBD–ACE2 complex structure was optimized based on the experimentally observed structure (PDB id: 6LZG). The obtained results were discussed by analyzing the 3D hydration shells of complexes and a trend for such thermodynamic quantities as SFE and PMF obtained by Equations (5) and (6) (see Section 4). The authors [44] studied the details of the 3D distributions of water oxygens and NAG nitrogen, as well as Na^+^ and Cl^−^ ions in solution around the SARS-CoV-2–hACE2 complex at different separations between proteins (0, 2, 4, 6, 8, and 10 Å). As found from these SDFs, irrespective of the protein separation, waters are oriented toward specific interaction sites of both proteins, while the NAG molecule is localized mostly in the area of the contact of protein surfaces, preferring to remain near the hACE2 receptor. The latter observation indicates the absence of any strong interaction between the drug-like molecule and the S protein. At the same time, Cl^−^ ions are located inside the hACE2 structure, and Na^+^ ions are predominantly arranged in the vicinity of the protein surfaces during their contact and separation (Figure 6). This detected localization of positively charged ions near the contacting surfaces of proteins in complex and water molecules near specific protein interaction sites is the result of, respectively, electrostatic interaction and H-bonding of the above solvent components with proteins. As was later established (see, for instance, [107]), these interactions play an essential role in the process of recognition of the hACE2 receptor by the virus spike protein and in stabilizing the RBD–hACE2 complex (see also the further review of the article [5]).

In addition to the designated interactions, hydrophobic interactions as well as salt bridges also contribute to binding affinity [107]. This aspect was also considered in [44]. First, for a series of protein separations, the authors [44] analyzed the 3D distribution of water oxygen atoms around selected pairs of amino acids of the S and hACE2 proteins (TYR83 from hACE2 and PHE486 from S as well as GLY326 from hACE2 and THR500 from S), which participate in hydrophobic interactions. The presented SDFs indicate almost full absence of the first hydration shell between interacting amino acids in the case of a separation of 0 Å (Figure 7a), which is evidence for an effect of excluded volume at short distances. At the same time, the hydration layer is clearly visible in SDFs at a separation of 4 Å (Figure 7b), that is an indication of water penetration into the cavity between the amino acid residues. The depleted region of water density at the surface was explained by the hydrophobic effect, i.e., the above hydrophobic residues are isolated from water due to the unfavorable solvation of nonpolar groups, thereby minimizing the surface exposed to solvent. Next, the authors calculated SFE and PMF for a series of separation distances (from 0 to 14 Å) between SARS-CoV-2 and hACE2. The obtained dependencies demonstrated the following features. SFE has a broad minimum (about –30 kcal/mol) at short separations (up to 4 Å), which is indicative of hydrophobic hydration playing an essential role in the protein–protein binding. At the same time, PMF exhibits a downslope shape with a deep minimum (less than −200 kcal/mol) at a separation of 0 Å, which reflects the intensity of binding between proteins.

SARS-CoV-2 virus entry into host cells is strongly dependent on the formation of RBD–hACE2 complexes. In the paper [5], using 3D-RISM and structural bioinformatics methods, the authors studied the binding and conformational properties of such complexes as well as the role of water-mediated interactions. The data obtained were compared with the corresponding results for the complex of the hACE2 receptor with the SARS-CoV virus, which was identified in 2003 and is also known as SARS-CoV-1. Because the complexes are too large, the authors of [5] only considered residues near the interface region that contribute significantly to binding. These residues were obtained from bioinformatics analysis. It should also be noted that, despite the known fact that ions play a significant role in the stabilization of protein complexes (see, for instance, Ref. [108]), in contrast to the article by Kobryn et al. discussed above [44], here the authors used a simplified model and treated the solvent as pure water. They aimed to identify essential structural differences that contribute to the higher transmissibility of SARS-CoV-2 relative to SARS-CoV, which is important for justifying the therapeutic targets that can prevent viral entry. To this end, the following S-protein cryo-EM structures, such as SARS-CoV (PDB id: 6CRZ), SARS-CoV-2 (PDB id: 6VYB), SARS-CoV-RBD–hACE2 complex (PDB id: 2AJF), and SARS-CoV-2-RBD–hACE2 complex (PDB id: 6M17), were analyzed. Bioinformatics analysis revealed the main characteristics of RBD binding to hACE2 and tested the effect of replacing SARS-CoV with SARS-CoV-2 on this binding. In particular, this analysis showed [5] that in the SARS-CoV-2, its RBD has expanded in size with a large conformational change compared to the case of SARS-CoV. Here, the RBD-up conformation of the SARS-CoV is less stable and more flexible, with less potential for binding with the receptor than in SARS-CoV-2. As a result, under RBD–hACE2 binding, the complex with CoV-2 becomes more rigid than the complex with CoV, “which favors proteolytic processing for membrane fusions” [5]. Three-dimensional RISM calculation of the binding energies for the complexes also showed that ∆Gbind is greater in the case of SARS-CoV-2-RBD–hACE2 in comparison with SARS-CoV-RBD–hACE2 (−57.2 kcal/mol vs. −50.1 kcal/mol, respectively).

The RBD-up conformation, binding with the hACE2 receptor, demonstrates stronger intermolecular interactions at the RBD–hACE2 interface “with differential distributions and the inclusion of specific H-bonds into the SARS-CoV-2-RBD-hACE2 complex”, as the authors emphasize [5]. These interfacial interactions are significant in the formation of the RBD–hACE2 complex. As follows from the 3D-RISM water distribution in the interfacial region of complexes (Figure 8), the interfacial water interacts strongly with both the RBD and hACE2 domains. However, the obtained data indicate stronger water-mediated interactions for the SARS-CoV-2–hACE2 complex than those for the SARS-CoV–hACE2 one.

Moreover, according to the 3D-RISM results, the interactions of the solvent with the complex are stronger than even with the pure virus. This effect is confirmed by the calculated protein–solvent interaction energies when the total interaction energy has a greater gain for the SARS-CoV-2–hACE2 complex due to the significant reduction in electrostatic (the Coulomb) energy (see Table 1 in Ref. [5] for details). The authors explained this effect by the stronger polarization of interfacial water in the case of the SARS-CoV-2–hACE2 complex. They also found that the SARS-CoV-2–hACE2 complex forms water bridges which are more stable in comparison with those of SARS-CoV–hACE2, which is confirmed by the values of 3D-RISM PMFs (Equation (5) in Section 4) between the RBD and hACE2 atoms (Figure 9). As one can see from Figure 9, a bridging water molecule is able to simultaneously form H-bonds with the Gln493 residue of SARS-CoV-2 and the Glu35 residue of hACE2. At the same time, for the SARS-CoV–hACE2 complex, the H-bond formed by bridging water is rather weak and diffusive (Figure 9). Thus, as the authors of [5] conclude, bridging water molecules may significantly contribute to the stabilization of the SARS-CoV-2–hACE2 complex.

Another study of the binding of the SARS-CoV-2 spike protein RBD to hACE2 and the role of solvation in this process was presented in [45]. The aim of the work was to trace changes in the structure and thermodynamics of the formed complex along the binding pathway using a combination of long-time-scale MD simulations and 3D-RISM theory. The MD method has been used to study the structural fluctuations of proteins upon binding. Respectively, the structures of the complex were extracted from the MD trajectories. According to these data, the RBD was only supposed to be a monomer in the up conformation. The 3D-RISM approach was used for calculation the hydration structure and thermodynamics, in particular, the SFE (Equation (6) in Section 4), the solvent-accessible surface area (SASA), and the PMV (Equation (7) in Section 4) for complexes in the 0.2 M NaCl aqueous solution at different states of protein–protein binding. 

Using umbrella sampling windows, to analyze the changes, the authors [45] identified three states of the complex: the initial, unbound state (window = 1), when the distance between the RBD and the hACE2 receptor is large (about 85 Å), indicating their independent hydration; the state characterizing the beginning of complex formation (window = 50), when the RBD–hACE2 interface is rather close (the interprotein distance is around 50 Å), so that the structure is similar to the bound state; and the bound state (window = 89) with the interprotein distance of about 48 Å. Figure 10 and Figure 11 demonstrate the 3D-RISM distribution of water oxygens and sodium and chloride ions around complexes in the unbound (window = 1) and bound (window = 89) state as examples.

As follows from Figure 10, in the unbound state of the complex, the water molecules are widely distributed on the surface of the RBD and hACE2 domains. One can also see the coordination of Na^+^ ions with the GLU484 and GLU471 residues of RBD. For chloride ions, there is a possibility for coordination with both proteins, namely, with LYS31 and LYS68 residues of hACE2, as well as with LYS452 and LYS458 residues of RBD. In the case of window = 50 (not shown here; for details, see the original paper [45]), there is enough distribution of solvent at the interface, but ion coordination is changed in strength and ion localization. Now, Na^+^ ions are arranged in the vicinity of GLU35 and ASP38 residues of hACE2, while there is no distribution of Cl^−^ ions near to the LYS31 residue of hACE2, which characterizes the weakened chloride coordination. In the bound state of the complex (Figure 11), the RBD–ACE2 interface is closely contacted with the formation of protein–protein H-bonds. In this case, some solvent distributions are saved at the interface, in particular, in the vicinity of the LYS31 residue of hACE2 and GLN493 and GLN498 residues of RBD. The strong interaction of Na^+^ is retained only with the GLU35 residue of hACE2, while with the ASP38 residue, it is significantly decreased. At the same time, in the bound state of the complex, the peaks characterizing interactions with Cl^−^ are very rare at the RBD–ACE2 interface.

Based on the obtained results, the authors [45] established the link between the changes in protein structures and thermodynamics upon complex binding. They proposed to consider the binding process with two separate stages, such as “a protein–protein approaching step and a local structural rearrangement step” [45]. In the first step, the energy of RBD–ACE2 interaction decreased when these proteins approached, but the SFE values increased. When the protein–protein distance was reduced to approximately 55 Å, H-bonds between RBD and ACE2 started to form. Upon further reduction of the distance, the amount of these H-bonds grew fast, and then about 6–10 H-bonds were found in the bound state of the complex. At the same time, when the RBD–ACE2 interface tends to come into close contact, proteins become dehydrated, and SFE values increase. As found by the authors on the basis of SASA and PMV analysis [45], the increase in SFE is determined mainly by the contribution of dehydration of hydrophilic amino acid residues on the protein surface. In addition, in support of the results in [5], it was shown that the solvent molecules forming H-bonds at the binding interface are bridging. As for the conformational changes of proteins, they were found only for the hACE2 receptor, namely, as the closing of its hydrophobic active-site pocket under binding with the RBD.

In addition to studying the features of SARS-CoV-2–hACE2 binding, the binding affinity of RBD to hACE2, and the role of the solvent in this process discussed above, the 3D-RISM method can also be used to study drug–virus interactions. Typical focuses of this field are the analysis of allosteric inhibition produced by ligands blocking sites involved in the RBD–hACE2 interaction or the analysis of the structural stability of the S protein and the target proteins of hACE2 in the presence of ligands with different binding properties to each of these two proteins [44]. This is especially important given the large number of SARS-CoV-2 mutations, so pharmacological inhibition of the RBD–hACE2 interaction is one of the most promising strategies for preventing virus replication in cells. As examples, one can see recent studies [109,110].

The purpose of the paper [109] was to evaluate the possibility of using FDA-approved marine drugs as inhibitors of SARS-CoV-2 main protease (M^pro^) on the basis of data from computationally aided drug-discovery and drug-development (CADD) methods such as field-template, QSAR (quantitative structure–activity relationship) [23], and molecular docking, as well as from the 3D-RISM approach. M^pro^, required for proteolytic processing of virus polyproteins and producing a number of components of the virus’s replication machinery, is a potential target in drug design. In this direction, the M^pro^ inhibitors can be considered as the perspective anticoronavirus agents (see, for instance, [111,112]). In the paper [109], three FDA-approved drugs derived from marine sources (Eribulin mesylate, Plitidepsin, and Trabectedin) were selected to examine their potential to be M^pro^ inhibitors.

The authors [109] have constructed the field-template and structure–activity atlas models to determine the molecular structure features of these inhibitors. They found that the above marine drugs have similar characteristics to already-known SARS-CoV-2 M^pro^ inhibitors. Moreover, the results of molecular docking have indicated that the inhibitors in question bind to the active-site pocket of M^pro^. Without dwelling in detail on the data obtained by CADD methods that can be found in the original article by Kalhotra et al., let us consider the results of the 3D-RISM technique since this tool and its possibilities are the main subject of this review. In their study [109], the authors used the 3D-RISM method to determine the solvation effects on the marine drug–M^pro^ binding. To this end, they analyzed the 3D distribution of water around the studied ligand–M^pro^ complexes (Figure 12) and found a high density of water formed at the active-site pocket of M^pro^ under binding. 

Moreover, only a part of the water molecules (oxygen distribution in green in Figure 12) favors the interactions between the drugs and M^pro^. Such molecules are difficult to displace by marine drugs, unlike molecules (oxygen distribution in red in Figure 12) that do not favor the protein–ligand interaction, and thus, the latter can be replaced by inhibitors. Following the authors’ reasoning based on the SDFs (Figure 12), the formed drug–M^pro^ complexes are stable in the presence of bulk water. The last statement is likely true, since it is a well-known fact that solvation contributes to the stability of the structure of both proteins and their complexes.

The same description of solvation effects on SARS-CoV-2—drug binding and conclusions from it—can be found in a paper by Madishetti et al. [110], where the discussion of the 3D-RISM results is essentially a copy of the discussion from the article [109] reviewed above. However, in both of the above papers, the authors’ conclusions about stability and about favorable/unfavorable interactions with water are speculative, since they can only be obtained from the calculation of solvation thermodynamics, i.e., from PMFs, SFE, or ΔG_bind_, but not based on solvent distribution analysis.

It should be noted that the solvation thermodynamics and structural maps of SARS-CoV-2 targets obtained by various computational techniques, including the 3D-RISM theory, are also presented in the special online repository [113]. As pushed in [113], this mapping is actively used in drug–protein interaction studies and is included in the rational drug design workflow of large pharmaceutical companies.

## 4. Methodology

Since the foundations of the molecular theory of solvation have already been described in detail in the literature [16,18,71], we give here briefly only some important aspects of its methodology that are relevant to the aim of our review. In the framework of the 3D-RISM method, the solvation structure is represented by a distribution function that determines the probability density of finding the interacting site of solvent molecules in a certain position in space around the solute, i.e., the 3D-RISM approach deals with the molecule–atom spatial distribution functions (SDFs), $g_{\beta }\left(r\right)$ ≡ $g_{\beta }(r,\Omega )$, for solvent sites (or atoms) *β* around the reference entire solute molecule. These functions are the three-dimensional density distribution functions of solvent atoms in a local coordinate system linked with the solute. Note also that the 3D-RISM equations operate on a system at a constant atomic density, and so the distribution functions, as well as the characteristics derived from them, correspond to the NVT ensemble.

To obtain the SDF, one fixes a solute at the origin of a local (spherical) reference frame and characterizes the local atomic densities by computing both the radial *r* and angular Ω=(θ,ϕ) coordinates of the site–site distance vector r. The SDF is determined from the 3D-RISM Ornstein–Zernike integral equation [73]:(1)$$h_{\beta }\left(r\right)=\sum_{\alpha }\int dr^{\prime }c_{\alpha }\left(r-r^{\prime }\right)\chi_{\alpha \beta }(r^{\prime })$$In this formula, $h_{\beta }\left(r\right)$ and $c_{\beta }\left(r\right)$ are the 3D total and direct correlation functions of solvent site β around the solute molecule, respectively, and $\chi_{\alpha \beta }(r)$ is the site–site solvent susceptibility. Prior knowledge about the bulk solvent susceptibility function is required; therefore, it is taken as input from the 1D-RISM theory. The indices *α* and *β* list all interaction sites for all types of solvent species. In addition, the relation between SDF and total is $g_{\beta }\left(r\right)=h_{\beta }\left(r\right)+1$. Note also that the 3D direct correlation function has asymptotic behavior such as $c_{\beta }\left(r\right)\;\sim -U_{\beta }\left(r\right)\;/\;k_{B}T$, where $U_{\beta }\left(r\right)$ is the 3D interaction potential between the whole solute molecule and solvent interaction site *β*, *k*_B_ is the Boltzmann constant, and *T* is temperature.

To solve the 3D-RISM equation, it must be complemented with a closure connecting also the 3D total and direct correlation functions. In general, the closure for Equation (1) can be written as [74]:(2)$$g_{\beta }\left(r\right)=\exp[-U_{\beta }\left(r\right)\;/\;k_{B}T+h_{\beta }\left(r\right)-c_{\beta }\left(r\right)+b_{\beta }\left(r\right)]\;$$

To date, around a dozen closures have been proposed for the Ornstein–Zernike-type integral equations, which differ from each other in the mathematical formulation of the bridge functional (or bridge function), $b_{\beta }\left(r\right)$. However, in studies of biomolecular solvation, the 3D Kovalenko–Hirata (KH) closure [75,76] is predominantly used for Equation (1) to obtain the SDFs: (3)$$g_{\beta }\left(r\right)=\left\{\begin{array}{cc} \exp\left(-U_{\beta }\left(r\right)\;/\;k_{B}T+h_{\beta }\left(r\right)-c_{\beta }\left(r\right)\right), & g_{\beta }\left(r\right)\leq 0\\ 1-U_{\beta }\left(r\right)\;/\;k_{B}T+h_{\beta }\left(r\right)-c_{\beta }\left(r\right), & g_{\beta }\left(r\right)>0 \end{array}\right.’$$The 3D-KH closure (3) is a coupling of the mean spherical approximation (MSA) for the regions of density enrichment ($g_{\beta }\left(r\right)>0$) with the hypernetted chain approximation (HNC) for the region of density depletion ($g_{\beta }\left(r\right)<0$). This closure is among the best closure relations to date in terms of both numerical stability and reasonable accuracy [77,78]. An important advantage of the KH closure is that it provides reliable numerical convergence of the 3D-RISM equations. This feature of KH closure plays an important role in many applications of 3D-RISM theory. Nevertheless, a well-known drawback of KH closure is the underestimation of the height of associative peaks of the 3D site distribution functions [39,59] due to the applied MSA in Equation (3) [59,71]. These peaks characterize the interactions of anions and negatively charged solute groups with water, contributing to the solvation process, which is very significant for biosystems. Despite the above drawback, the 3D-KH closure relation is able to properly account for the solvation structure of complex solvated systems with significant association effects [78], providing accurate average solvation characteristics, when their values change insignificantly in comparison with the data of direct structural experiments and MD simulations. This problem with peak underestimation can be overcome in other ways. One could apply more sophisticated approaches such as the site density functional theory (SDFT) or the cluster site density functional theory (CSDFT) [20], but for the moment, they can be hard to work with biomacromolecules due to large computational expenses. Another way is to use the correction for Equation (2) by a bridge function or bridge functional. However, there is no universal form for $b_{\beta }\left(r\right)$, so it must be carefully selected for each specific biosystem, which already poses a significant difficulty. It should also be noted that the 3D-HNC approximation greatly overestimates the association, and therefore, the 3D-RISM-HNC equations often diverge for the cases of electrolyte solutions or macromolecules in polar solvents [39,59]. Other known closures (Percus–Jevick, Martynov–Sarkisov, Ballone–Pastore–Galli–Gacillo, etc.) do not take into account the electrostatic asymptotic behavior of the interaction potential in a proper manner [59,78].

The numerical solution of the 3D-RISM Ornstein–Zernike (OZ) equation with the appropriate closure relation yields the SDFs which are often presented in terms of isodensity surfaces at some probability level, for instance, relative to bulk system. From these functions, information on the solvation structure and solvation thermodynamics is extracted. For instance, using SDFs, one can obtain the following:

* Visualization of the hydration layer around the solute molecule at various thresholds of isodensity surfaces, such as visualization of the whole hydration layer around the molecule, the localized hydration shell only near the hydrophilic parts of the molecule surface with characterization of possible H-bonding, the distribution of solvent molecules in the nearest environment of the molecule functional groups with characterization also of possible H-bonding, or the localization of inner water molecules, which are strongly bound to the biomolecule in the case of proteins.

* The total hydration number of the solute molecule,
(4)$$n_{\mathrm{tot}}=\rho_{O_{W}}\int_{V_{\mathrm{hs}}}g_{O_{W}}\left(r\right)dV$$
where *V*_hs_ is the volume occupied by the first hydration shell of the entire solute molecule, and *ρ*_Ow_ is the average number density of water oxygen atoms. 

* The potential of mean force (PMF),
(5)$$W_{\beta }\left(r\right)=-k_{B}T\;{\ln[g}_{\beta }\left(r\right)]$$
which is an energetic characteristic representing the ratio of the free energy of a solvent particle at a given distance from the solute to that in the bulk. Or, in other words, it allows the site binding affinity of solvent species to the solute molecule to be estimated [59]. For biochemistry, biophysics, and pharmaceuticals, this represents a new approach to molecular recognition and computational fragment-based drug design (see, for instance, [36,79]).

* Solvation free energy (SFE) of the solute in multicomponent solvent.

In particular, in the framework of 3D-RISM-KH approach, this energy is expressed in terms of the 3D solvent site distribution functions as [71]
(6)$$∆G^{solv}(KH)=k_{B}T\sum_{\beta }\rho_{\beta }\int dr\left\{\frac{1}{2}h_{\beta }(r{)}^{2}\Theta [-h_{\beta }(r)]-c_{\beta }(r)-\frac{1}{2}h_{\beta }(r)c_{\beta }(r)\right\}$$
where Θ(x) is the Heaviside step function. 

* Partial molar volume (PMV) and compressibility.

The PMV of the solute molecule at the infinite dilution limit, obtained in the framework of the Kirkwood–Buff theory in terms of the 3D-RISM direct correlation functions, is calculated as [80,81]
(7)$$\bar{V}=k_{B}T\chi_{T}\left(1-\sum_{\beta }\rho_{\beta }\int drc_{\beta }(r)\right)$$
where $\chi_{T}$ is the isothermal compressibility of the bulk solvent, which is expressed in terms of the site–site (1D) direct correlation functions of the bulk solvent [82] by
(8)$$\rho k_{B}T\chi_{T}={\left(1-4\pi \sum_{\alpha \beta }\rho_{\alpha }\int_{0}^{\infty }r^{2}drc_{\alpha \beta }(r)\right)}^{-1}$$
where $\rho =\sum_{i}\rho_{i}$ is the total number density of the bulk solvent mixture of molecular species *i*.

## 5. Conclusions

Viral entry into the host cell, subsequent replication, and transmissibility are essentially dependent on receptor recognition, binding, and membrane fusion mechanisms. One of the non-empirical methods that makes it possible to investigate at the molecular level the mechanism of virus penetration through the cell membrane and the ways of its inhibition is the 3D-RISM method based on statistical mechanics treatment. The presented review shows that this approach is capable of properly studying solvation effects and their role in protein–protein binding. In addition, the 3D-RISM technique provides the ability to analyze the action of ligand molecules, and, thus, it is resourceful for accurate screening of potential drugs. 

The viruses discussed in this review are so huge that they cannot be simulated at the atomistic level, taking into account all the molecular details. On the other hand, for proper calculations of virus activity or functionality, we have to account for the real conformation of inhibitor–virus or membrane protein–virus complexes. Such information is not known a priori in many cases. As a result, the 3D-RISM calculations are to be added by methods providing self-consistent evaluations not only the liquid environment but also the molecular structure of the virus complexes. The latter can be carried out in two ways, i.e., by combining the 3D-RISM approach with molecular mechanics [43,98] or with bioinformatics analysis [4,5]. The first way requires essential computational resources; typically, the time of calculations rises more than two orders. The combination of the 3D-RISM with bioinformatics tools assumes the use of extensive data sets. Despite recent progress in virtual screening and machine learning for virus–ligand complexes (see, for instance, [114,115,116,117]), information about the actual conformation of hydrated virus complexes is rather limited so far. However, we can make the following statement based on our own results [5,37,40] and comparative analysis of data from various computational methods [118] for proteins and solvated protein–ligand complexes. Namely, 3D-RISM calculations are quite accurate in determining protein and protein–ligand hydration structure, including the localization of water molecules within their structure (internal water molecules) or at the protein interface. At the same time, the 3D-RISM treatment of energetic contributions to the binding free energy of the virus–ligand complex requires a caution, since it is rather sensitive to conformational details. Nevertheless, the presented examples demonstrate that the 3D-RISM approach is promising for such evaluations. From a fundamental point of view, the results discussed above contribute to a deeper understanding of the features of functioning of viruses, which is necessary for an adequate and complete fight against viral infections. In the context of the latter, being more accurate than molecular docking and faster than MD simulations, 3D-RISM theory could be useful for studying the effects of viral mutations found in new strains around the world.

## Figures and Tables

**Figure 1 ijms-25-03697-f001:**
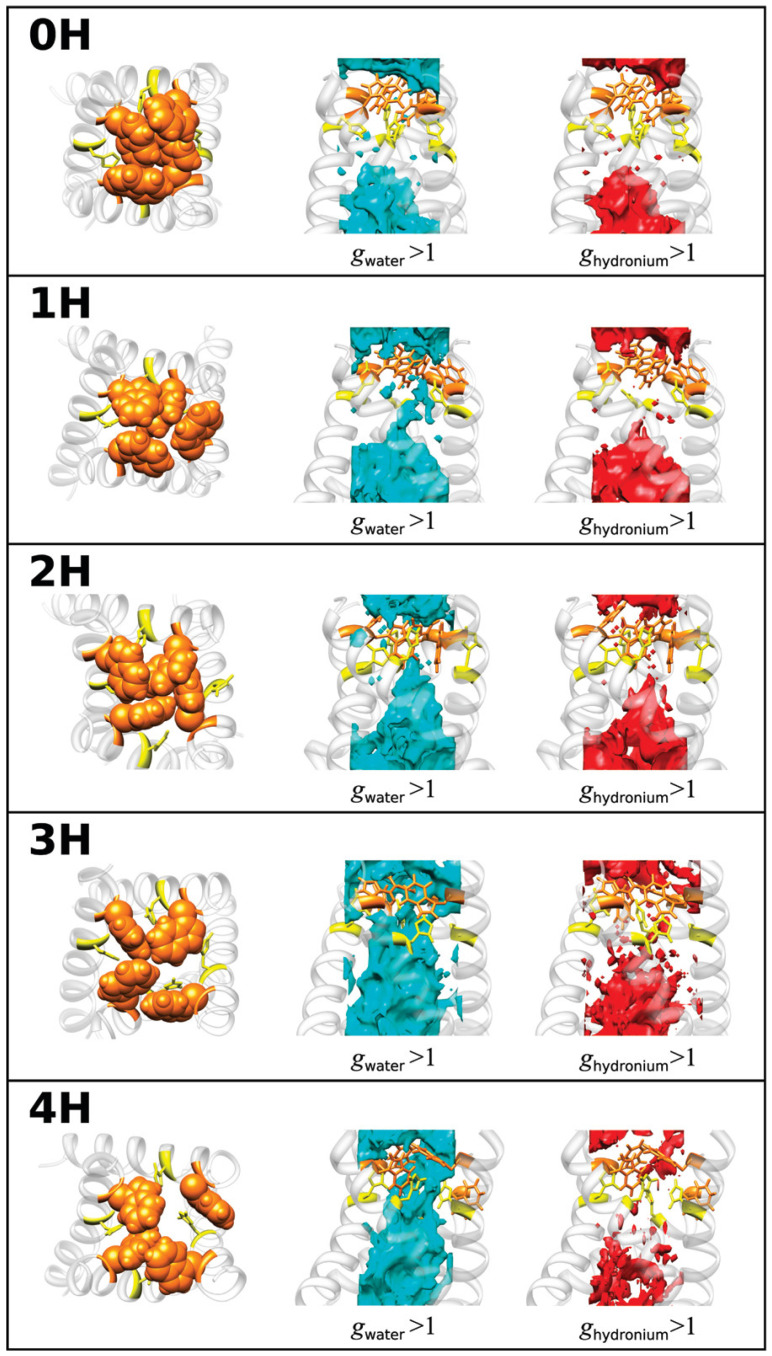
SDFs of water in the channel (cyan) and hydronium ion (red), with g(**r**) > 1. The molecular structure of the channel gating region is also depicted on the left for protonation different states of the His37: Trp41, orange; His37, yellow. Reprinted (adapted) with permission from {Phongphanphanee S., et al. J. Am. Chem. Soc. 2010, 132, 28, 9782–9788 [41]}. Copyright {2010} American Chemical Society.

**Figure 2 ijms-25-03697-f002:**
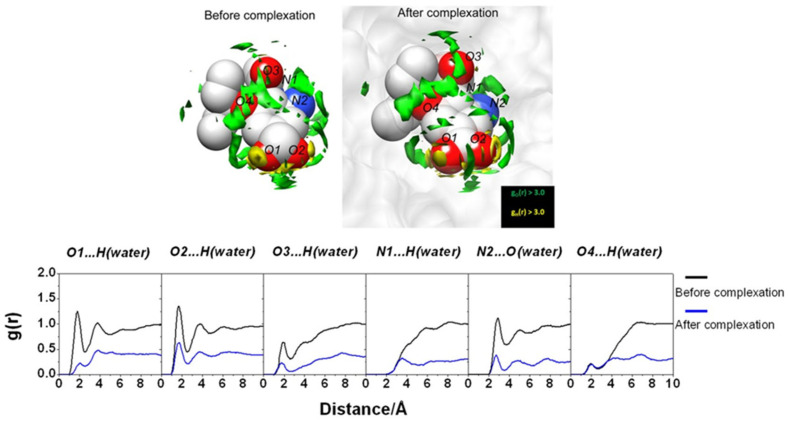
(**Upper**) SDF of water O atom and H atom around oseltamivir via 3D-RISM calculation with g(**r**) > 3.0 before and after the complexation. (**Lower**) The 3D-RISM radial distribution functions of hydrogen-bonded pairs between oseltamivir heteroatoms and water molecules before and after the complexation calculated from the free oseltamivir and oseltamivir in complex with wild-type neuraminidase. Reprinted from Phanich et al. [43], Copyright (2016), with permission from John Wiley & Sons, Inc.

**Figure 4 ijms-25-03697-f004:**
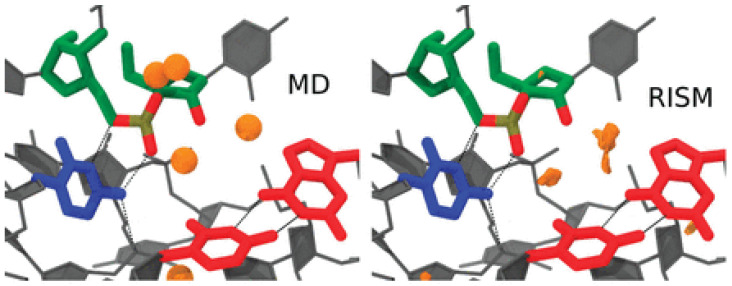
Na^+^ distribution functions (orange isosurfaces) from 3D-RISM calculations compared with peak positions (orange spheres) from MD simulations. Isosurfaces correspond to a concentration of 300 times the bulk (140 mM). Density more than 3 Å from the active-site residues was clipped for clarity. Adapted from Radak et al. [42], Copyright (2015), with permission from Cold Spring Harbor Laboratory Press and the RNA Society.

**Figure 5 ijms-25-03697-f005:**
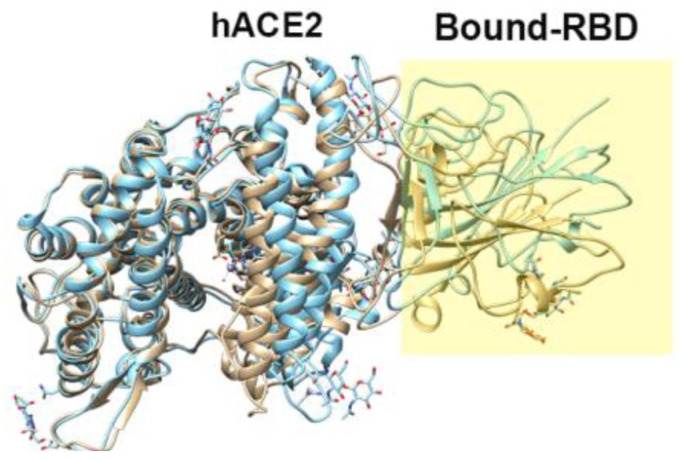
Example of the structure of the SARS-CoV-2-RBD–hACE2 complex (PDB id: 6M17).

**Figure 6 ijms-25-03697-f006:**
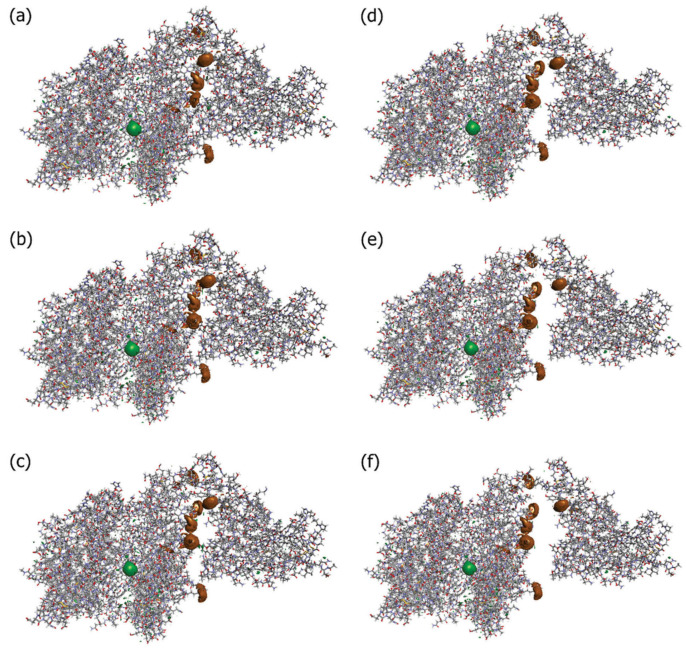
Isosurfaces of the 3D distribution of Na^+^ (brown) and Cl^−^ (green) ions in solution around proteins SARS-CoV-2 and ACE2 (sticks) at a series of separations between them: (**a**) corresponding to the 0 Å distance, (**b**) 2 Å, (**c**) 4 Å, (**d**) 6 Å, (**e**) 8 Å, and (**f**) 10 Å. For both cases of ions, the isosurface values are 6.5. These values are close to their maximum for the current 3D-RISM solution, which means that they show the most probable positions of the respective solvent sites around the proteins. Reproduced from Kobryn et al. [44], Copyright (2021), with permission from the Centre National de la Recherche Scientifique (CNRS) and the Royal Society of Chemistry.

**Figure 7 ijms-25-03697-f007:**
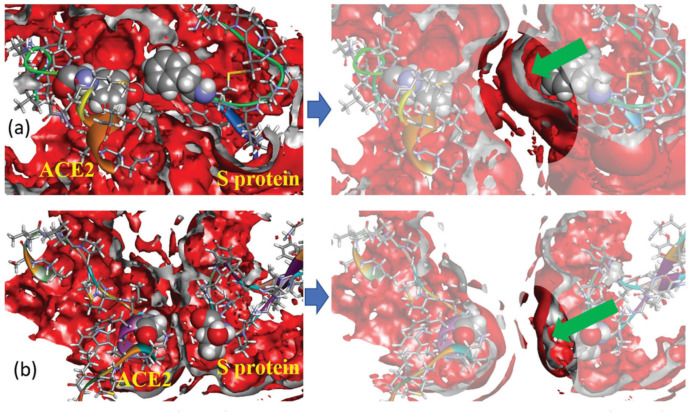
Three-dimensional distribution of water oxygens (rear view) around hydrophobic fragments of hACE2 and a spike of SARS-CoV-2 at two separations of proteins: 0 Å (left column) and 4 Å (right column); the isosurface value is 1.3 in all cases (this corresponds to the first solvation shell); the isosurface color is red; the inner part of the isosurface is gray. Protein fragments are drawn as ribbon sketches and sticks; selected amino acid residues are displayed as van der Waals surfaces. Subfigures on top (**a**) show the 3D distribution for the pair of amino acids TYR83 (hACE2) and PHE486 (spike), while the subfigures at the bottom (**b**) show the 3D distribution for the pair GLY326 (hACE2) and THR500 (spike). Green arrows in the right part of the figure direct into a position of first solvation peak around residues. Reproduced from Kobryn et al. [44], Copyright (2021), with permission from the Centre National de la Recherche Scientifique (CNRS) and the Royal Society of Chemistry.

**Figure 8 ijms-25-03697-f008:**
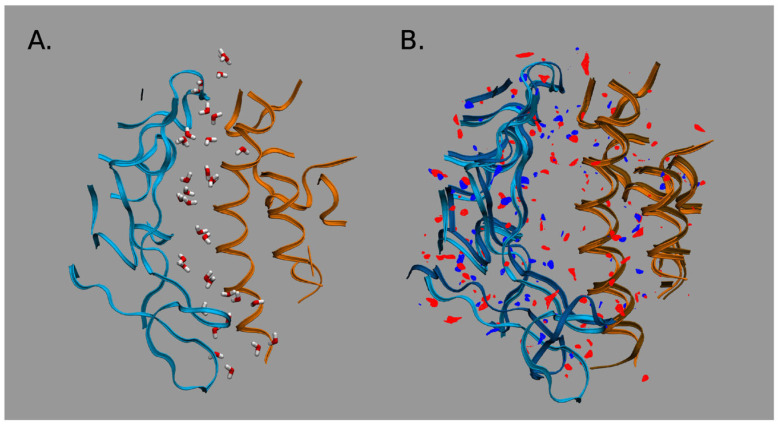
Water distribution in the interfacial region of complexes: (**A**) For the CoV-2–hACE2 complex. (**B**) Differences in distributions of water oxygens (blue color) and water hydrogens (red color) between the CoV-2–hACE2 and CoV–hACE2 complexes. The RBD is indicated by blue ribbons, the hACE2 by orange ribbons, and the CoV–hACE2 is shown as background for the differences. Reproduced from Kumawat et al. [5], Copyright (2022), with permission from MDPI.

**Figure 9 ijms-25-03697-f009:**
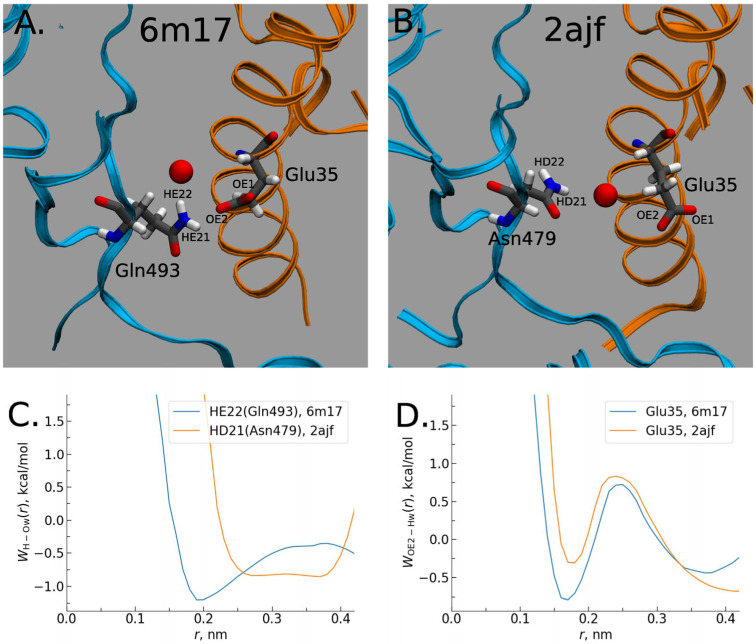
Water bridging the complexes: (**A**) Location of water oxygen bridging the CoV-2–hACE2 complex. (**B**) The same for the CoV–hACE2 complex. (**C**) PMF of H-O distribution for the CoV-2 and CoV water bridges. (**D**) PMF of O-H distribution for the hACE2 water bridges in the complexes. Reproduced from Kumawat et al. [5], Copyright (2022), with permission from MDPI.

**Figure 10 ijms-25-03697-f010:**
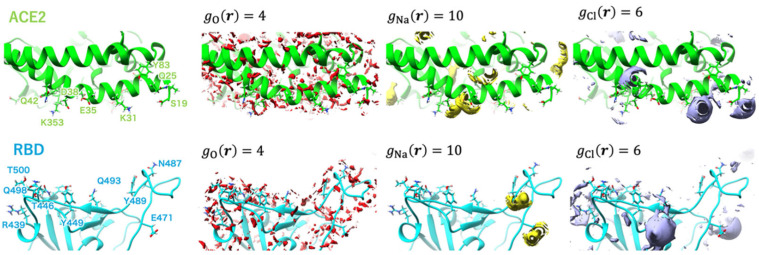
Isosurfaces of the SDFs of water oxygen, sodium ion, and chloride ion at window = 1 depicted in red-, yellow-, and purple-colored surfaces, respectively. The labels of the major amino acid residues involved in interprotein hydrogen bonding are shown in the leftmost panel. Reprinted (adapted) with permission from {N. Yoshida, et al., J. Chem. Inf. Model. 2022, 62, 11, 2889–2898 [45]}. Copyright {2022} American Chemical Society.

**Figure 11 ijms-25-03697-f011:**
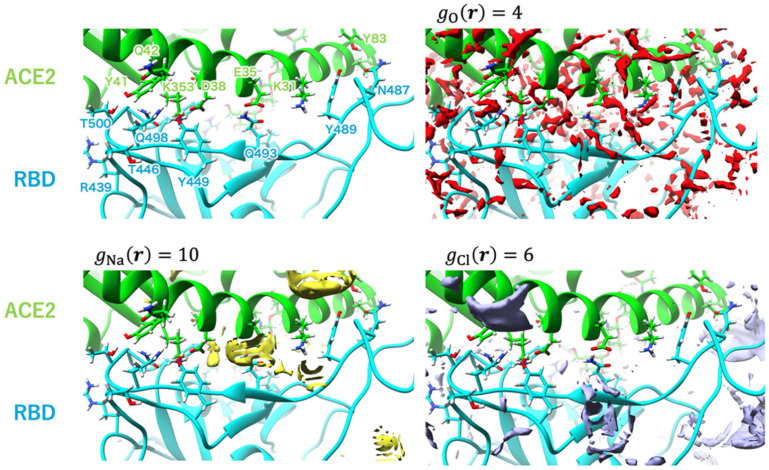
Isosurfaces of the SDFs of water oxygen, sodium ion, and chloride ion at window = 89 depicted in red-, yellow-, and purple-colored surfaces, respectively. The labels of the major amino acid residues involved in interprotein hydrogen bonding are shown in the upper left panel. Reprinted (adapted) with permission from {N. Yoshida, et al., J. Chem. Inf. Model. 2022, 62, 11, 2889–2898 [45]}. Copyright {2022} American Chemical Society.

**Figure 12 ijms-25-03697-f012:**
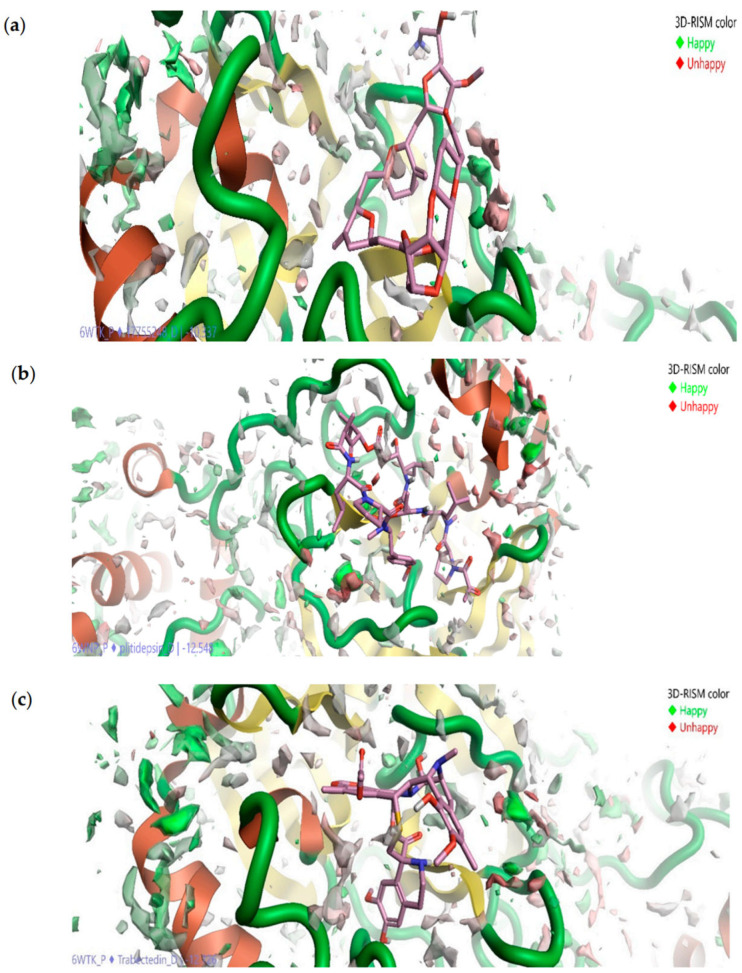
Three-dimensional distribution of water oxygens around ligand–protein complex: Eribulin mesylate–M^pro^ (**a**), Plitidepsin–M^pro^ (**b**), and Trabectedin–M^pro^ (**c**). The oxygen distribution in green and red colors indicates, respectively, favorable and unfavorable location of water at the active site of M^pro^. Reproduced from Kalhotra et al. [109], Copyright (2021), with permission from MDPI.

## Data Availability

Not applicable.

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
