# Peer review of "The Three-Dimensional Reference Interaction Site Model Approach as a Promising Tool for Studying Hydrated Viruses and Their Complexes with Ligands"

_ijms, 2024, doi:10.3390/ijms25073697_

Round 1
Reviewer 1 Report
Comments and Suggestions for Authors
The manuscript addresses a highly relevant topic for the scientific community. From my perspective, the manuscript can be published after minor revisions.
-Understanding the molecular mechanisms involved in viral invasion processes is of utmost importance for the development of effective treatments against diseases. However, few studies have been presented using 3D-RISM for a wider variety of viruses. The question is whether there is room to further explore the literature review or if there are indeed few works available.
- The authors should improve the quality of the images, which is currently quite low.
- I missed the comparison of the results with experimentally verified effects. Would it be possible to establish this relationship for some results?
- The authors did not comment on the limitations of the method. Since this is an article related to the method itself, it would be interesting to add this discussion.
- The authors could add a discussion extrapolating the current use of the method and showing its potential for future research.
- Which other biological components besides proteins could this method be used for? It would be interesting to add a section presenting these results related to viruses.
Comments on the Quality of English LanguageModerate editing of English language required
Author Response
Dear Reviewers,
thank you very much for careful and critical reading of the manuscript. Your remarks helped us to improve the text and to return the missing moments. We have tried to take into account all addressed points. The necessary changes in the manuscript are outlined by blue color.
Reviewer 1
1. Understanding the molecular mechanisms involved in viral invasion processes is of utmost importance for the development of effective treatments against diseases. However, few studies have been presented using 3D-RISM for a wider variety of viruses. The question is whether there is room to further explore the literature review or if there are indeed few works available.
The reviewer is right that there are currently only a few papers on the viruses using the 3D-RISM approach which are discussed in our manuscript. We tried to cover all these papers.
2. The authors should improve the quality of the images, which is currently quite low.
In the manuscript, the figures have been replaced with figures of sufficiently high resolution.
3. I missed the comparison of the results with experimentally verified effects. Would it be possible to establish this relationship for some results?
This manuscript is a review of existing literature on the application of the 3D-RISM method to the study of viruses. Accordingly, we discussed primarily these results and the possibilities of 3D-RISM approach to this type of research. If the original sources did not contain a comparison with experimental data, then, of course, we did not provide it. Nevertheless, it should be noted that such subtle effects as the structural features of the hydration of viruses and their complex formation with drugs are not yet available for structural experiments. However, the initial crystal structures of the complexes are taken from the corresponding experimental data from the PDB bank, and these structures are well reproduced by 3D-RISM approach (see, e.g., Molecules 2022, 27, 799; Int. J. Mol. Sci. 2022, 23, 14785; J. Mol. Liq. 2023, 384, 122281.)
4. The authors did not comment on the limitations of the method. Since this is an article related to the method itself, it would be interesting to add this discussion.
These limitations are discussed in Methodology (Section 4) in original manuscript.
5. The authors could add a discussion extrapolating the current use of the method and showing its potential for future research.
The potential of 3D-RISM approach for future research is noted in Conclusions (Section 5) in original manuscript.
6. Which other biological components besides proteins could this method be used for? It would be interesting to add a section presenting these results related to viruses.
The 3D-RISM method can be applied to various biocompounds, as specified in Section 2 (2. 3D-RISM approach as a promising tool for studying hydrated biomolecules). Section 3 is directly devoted to the study of viruses (3. Applications of 3D-RISM theory for studying viruses)
7. Comments on the Quality of English Language
Moderate editing of English language required
English language was improved in the text of manuscript.
We hope so much that the reviewers will be satisfied with the corrections and additions.
Reviewer 2 Report
Comments and Suggestions for Authors
The manuscript “3D-RISM Approach as a Promising Tool for Studying Hydrated Viruses and Their Complexes with Ligands” is devoted to the application of 3D-RISM approach for the investigation of hydrated viruses and virus complexes. I believe the 3D-RISM approach is one of perspective liquid-state theories based on statistical mechanics. The advantages of the theory is its computational effectiveness, with the down side of being an approximate approach.
The manuscript is written as a mini-review focusing on the applications of the method to viruses or protein-ligand complexes. In general, I do not see serious gaps in this manuscript, however some minor details need to be improved:
1) Appropriate references to the relevant reviews should be added when the authors describe well-documented methods PB (line 233), GBSA (line234), and QSAR (line 543).
2) the symbol Χαβ (line 604) should be corrected.
3) the symbol r2 should be removed in the integrand function (8) or bold symbols r to be replaced by normal font.
Author Response
Dear Reviewers,
thank you very much for careful and critical reading of the manuscript. Your remarks helped us to improve the text and to return the missing moments. We have tried to take into account all addressed points. The necessary changes in the manuscript are outlined by blue color.
Reviewer 2
1) Appropriate references to the relevant reviews should be added when the authors describe well-documented methods PB (line 233), GBSA (line234), and QSAR (line 543).
Appropriate references to relevant reviews were added to the text and list of references.
Ref [21] for PB in line 235; Ref [22] for GBSA in line 236; Ref [23] for QSAR in line 549.
2) the symbol Χ_αβ (line 604) should be corrected.
This symbol was corrected (now it's line 608).
3) the symbol r2 should be removed in the integrand function (8) or bold symbols r to be replaced by normal font.
The symbol r in formula (8) was replaced with the appropriate font.
We hope so much that the reviewers will be satisfied with the corrections and additions.